# Multimodal Intraoperative Image-Driven Surgery for Skull Base Chordomas and Chondrosarcomas

**DOI:** 10.3390/cancers14040966

**Published:** 2022-02-15

**Authors:** Walid I. Essayed, Parikshit Juvekar, Joshua D. Bernstock, Marcio S. Rassi, Kaith Almefty, Amir Arsalan Zamani, Alexandra J. Golby, Ossama Al-Mefty

**Affiliations:** 1Department of Neurosurgery, Brigham and Women’s Hospital, Harvard Medical School, Boston, MA 02115, USA; wibnessayed@bwh.harvard.edu (W.I.E.); pjuvekar@bwh.harvard.edu (P.J.); jbernstock@bwh.harvard.edu (J.D.B.); agolby@bwh.harvard.edu (A.J.G.); 2Department of Neurosurgery, A. C. Camargo Cancer Center, São Paulo 01509-001, Brazil; marcio.rassi@accamargo.org.br; 3Department of Neurosurgery, Barrow Neurological Institute, St. Joseph’s Hospital and Medical Center, Phoenix, AZ 85013, USA; kaith.almefty@barrowbrainandspine.com; 4Department of Radiology, Brigham and Women’s Hospital, Harvard Medical School, Boston, MA 02115, USA; azamani@bwh.harvard.edu

**Keywords:** chordomas, chondrosarcoma, intraoperative MRI, multimodal imaging, neuronavigation, resection, skull base

## Abstract

**Simple Summary:**

Achieving gross total resection during the first surgical intervention is particularly important for chondrosarcomas and chordomas, as recurrences are frequently impossible to resect due to post-surgical and post-radiation scarring and vascular fragility. Despite overall survival and progression-free survival being strongly dictated by gross total resection, it is reportedly achieved in less than 70% of patients. While the individual utility of several imaging modalities such as intraoperative CT, MRI, ultrasound, endoscopy, fluoroscopy and neuronavigation has already been demonstrated in previous literature; our case series highlights the importance and methodology of their simultaneous, real-time integration in the Advanced Multimodality Image-Guided Operating (AMIGO) suite at our institution to maximize of resection and mitigate complications.

**Abstract:**

Given the difficulty and importance of achieving maximal resection in chordomas and chondrosarcomas, all available tools offered by modern neurosurgery are to be deployed for planning and resection of these complex lesions. As demonstrated by the review of our series of skull base chordoma and chondrosarcoma resections in the Advanced Multimodality Image-Guided Operating (AMIGO) suite, as well as by the recently published literature, we describe the use of advanced multimodality intraoperative imaging and neuronavigation as pivotal to successful radical resection of these skull base lesions while preventing and managing eventual complications.

## 1. Introduction

Recent advances in technology have enabled considerable progress in neurosurgical practice and associated clinical outcomes. This revolution has in part been driven by the ever-expanding armamentarium of imaging tools available in the operating room. In skull base surgery, widespread adoption and use of neuronavigation, endoscopes, and intraoperative imaging represent the core of such advances [1,2]. In line with such advances, our institution has developed the Advanced Multimodality Image-Guided Operating (AMIGO) suite in an effort to maximize the potential of modern imaging tools to guide challenging procedures [3]. The AMIGO suite consists of three interconnected rooms with a 3T intraoperative magnetic resonance imaging (MRI) scanner, a positron emission tomography (PET)/computed tomography (CT) scanner, endoscope(s), an endovascular fluoroscopy suite, ultrasounds, and a navigation system that is capable of integrating all aforementioned imaging modalities in real time (Figure 1) [3]. The power and potential of the multimodal approach in guiding surgeries for chordomas and chondrosarcomas are readily apparent. Radical resection plays a central role in the management of these tumors, and it is particularly critical to achieve this during the first surgical intervention [4]. Accordingly, herein we review our series of skull base chordoma and chondrosarcoma resections performed in the AMIGO suite in an effort to highlight the concept of multimodal-imaging driven skull base surgery.

## 2. Methods

Patients with recurrence of a known chordoma and chondrosarcoma or who were harboring new lesions consistent on imaging with chordoma and chondrosarcoma underwent prospective surgical resection in the AMIGO suite [3]. The surgeries were performed by the senior author (OA) at the Brigham and Women’s Hospital between January 2013 and December 2020. Under Institutional Review Board approval (IRB2008P001024), patients’ data, including age, sex, tumor (location, extension, volume), and prior treatment(s), were obtained/included. Data regarding preoperative imaging and planning, surgical approaches, intraoperative imaging, other implemented modalities, and surgical outcomes were also examined and included. All involved patients had histologically and immunohistochemistry confirmed chordoma or chondrosarcoma [5]. All patients agreed to surgical interventions and publication of their images.

### 2.1. Operative and Perioperative Management in the AMIGO Suite

All patients included within the study underwent a surgical procedure aimed at achieving maximal surgical resection of their skull base lesions. The surgical approach for each patient was determined after a thorough review of both the clinical presentation and preoperative imaging. All preoperative imaging was reviewed, co-registered, segmented, and uploaded to the clinical navigation system prior to surgical intervention. If deemed necessary, intraoperative fluoroscopy was used to place a lumbar drain for CSF drainage. Proper positioning is crucial to the ability to perform multimodality imaging (CT, MRI, Angiography) intraoperatively. Close coordination with the neuromonitoring and anesthesia teams is crucial in the setup, particularly in the iMRI setting. The groin was prepped for rapid femoral artery catheterization if the vascular risk was deemed high. 

When maximal resection was thought to have been achieved, the patient and surgical room were prepared according to the iMRI safety protocol. Specific sequences were obtained depending on the preoperative appearance of the tumor. Thin cut T2 and T1 without contrast are particularly helpful. MPRAGE T1 with contrast, with and without fat suppression, is also frequently employed. Once acquired, intraoperative imaging was reviewed and discussed by the neuroradiology and operative teams. Segmentation of the residual target volume was completed and transferred to the navigation system to help guide the remainder of surgery and additional resection if needed. In cases where the verification of adequate bony resection was needed, a CT scan was also obtained. A CT scan was also obtained as immediate postoperative imaging in selected cases. Preoperative and intraoperative segmentations and navigation were performed using a commercially available navigation system (Brainlab AG, Munich, Germany).

### 2.2. Volumetric Analysis

For the imaging analysis, a neurosurgeon and neuroradiologist completed the imaging segmentation and review retrospectively. Preoperative, intraoperative, postoperative, and follow-up imaging were co-registered. Postoperative imaging was defined as any MRI obtained within three months of surgical intervention, including intraoperative MRI when it demonstrated gross total resection. Postoperative residual lesions were labeled and segmented based on three-month follow-up imaging and/or given the presence of any suspicious findings that demonstrated progression on subsequent follow-up imaging. The intraoperative residual percentage was calculated based on the initial tumor volume and on the suspected residual volume on the iMRI. The postoperative residual percentage was based on the postoperative residual volume segmentation. Subtotal resection was defined as a resection >90% of initial tumor volume, whereas partial resection was defined for resection <90%. Volumetric analysis was performed using Slicer 3D (www.slicer.org. accessed on 22 June 2021) [6].

### 2.3. Clinical Outcomes and Follow-Up

Any events that required reoperation, extended inpatient care, and/or higher-acuity care were classified as major complications; other events were classified as minor complications. The surgical resection performed in AMIGO was considered the index surgery for the surgical follow-up evaluations. Surgical follow-up was calculated from the index surgery to the selected follow-up end date of 31 December 2020. The overall follow-up was calculated based on the initial chordoma diagnosis and on the last available follow-up or follow-up end date. The progression-free survival (PFS) was based on clinical deterioration or local radiological progression after the index intervention. All statistical analyses were performed using Microsoft Excel (Microsoft, Redmond, Washington, WA, USA) and SPSS version 23 (IBM Inc., Armonk, New York, NY, USA).

## 3. Results

### 3.1. Patient Population, Tumor Characteristics, and Preoperative Imaging

This cohort included nine patients (six females and three males) who underwent 11 surgeries in the AMIGO suite from January 2013 to December 2020 (Table 1). Three of the surgeries were performed on the same patient, with a three-year interval between each procedure. A total of six patients had chordomas, and three had chondrosarcomas. Tissue pathology revealed a conventional chordoma subtype in all chordoma patients. All three chondrosarcoma patients had grade I lesions, with two having been classified as a myxoid subtype.

Six of the eleven surgeries (54%) were performed on recurrent lesions, with an average of two prior surgeries before our index intervention. All three chondrosarcoma cases were newly diagnosed. All recurrent patients had previously undergone proton beam radiation, with one patient having undergone a combination of proton beam and gamma knife treatment (surgery 2, Table 1). The mean age at the time of the index surgery was 49 (±11.2) years. Six surgeries were performed for worsening constellation of clinical symptoms, and the remaining five were performed after radiological growth on interval imaging. Only one patient had a normal neurological exam prior to the surgery. The remaining eight patients (89%) had some form of neurological impairment preoperatively (i.e., primarily cranial nerve deficits). In line with the natural history of these tumors, all the lesions appeared to have originated at the clivus, with a cavernous and petrous extension in 72% of cases. An intradural extension was appreciated in 45% of the cases prior to the index surgery. All demographic, clinical, and tumor characteristics are summarized in Table 1 and Table 2.

### 3.2. Surgery and Intraoperative Imaging 

A diverse spectrum of surgical approaches was utilized in our cohort; the different approaches are summarized in Table 3. iMRI was performed for all eleven cases. Eight interventions (72%) involved using three or more imaging modalities. The fluoroscopy setup was used for the placement of a lumbar drain during five interventions. A preoperative femoral sheath was placed in two cases, and a cerebral angiogram was performed in one case. Endoscopy was used in ten procedures. iMRI showed residual tumors in seven surgeries, allowing further resections in six. Details regarding intraoperative imaging are summarized in Table 4. 

### 3.3. Volumetric Analysis

The average tumor volume was 13.4 cm^3^ (2.56–40.7 cm^3^). Surgical resection yielded a minimum of 89.4% debulking in our series. Gross total resection was achieved in three surgeries (27%), with four additional surgeries having achieved 98–99% resected tumor volumes. Notably, gross total resection was achieved in all of the three newly diagnosed tumors. When further resection was possible after using iMRI, the resection volume was extended on average from 92.8% to 97.7%. Finally, there was no statistical difference between the overall extent of resection of chordomas versus chondrosarcomas in our AMIGO cohort (Table 5).

### 3.4. Clinical Outcomes and Follow-Up

With regard to the six surgeries performed for worsening symptoms, clinical improvement was obtained in two patients, and the remaining four patients had stable postoperative symptoms and neurological exam findings. No mortality was recorded in our AMIGO series. One significant event of note was encountered during the third recurrence intervention in patient # 3, with the patient having suffered an intraoperative intracavernous carotid injury that was successfully repaired with a Sundt clip graft, with a reassuring intraoperative angiogram and without subsequent sequalae. No CSF leaks were recorded in our series. A total of five patients had minor complications, with four patients having developed persistent partial trigeminal and frontalis nerve deficits and one patient having a postoperative deep vein thrombosis.

The average postoperative follow-up in our series was 2.9 years (0.3–7.2 years), with only one death during the follow-up having been recorded. The average overall survival was 2.8 years (0.3–7.2 years). The overall follow-up since initial diagnosis was 11.48 years (±8 years). 

The two patients with newly diagnosed chordomas underwent proton beam radiation after their index surgeries. One patient developed irradiation-induced panhypopituitarism. A total of two patients developed drop metastases in the cervical and lumbar spine, requiring additional radiation and surgery. The disease continued to progress in one of these patients, requiring salvage treatments including imatinib and erlotinib, which were unsuccessful. This patient succumbed after a total of 22 years from his initial diagnosis, representing the only death in this series.

### 3.5. Case Examples

#### 3.5.1. Case 1

A 26-year-old, otherwise healthy female (patient # 9) presented with progressively worsening double vision. Her neurological examination was notable for a complete right sixth nerve palsy. Her MRI demonstrated an enhancing lesion of the upper clivus with a focal area of dural penetration abutting the basilar artery. The computed tomography (CT) scan demonstrated a partially calcified bony lesion with partial erosion of the right posterior clinoidal process (PCP) (Figure 2). The computed tomography angiography (CTA) scan confirmed the close relationship of the tumor to the basilar tip and its terminal branches, as well as to the anterior displacement of the right cavernous internal carotid artery (ICA) (Figure 2). Three-dimensional modeling of the tumor within the skull base made it apparent that the tumor would not be well exposed via a middle fossa approach, thereby guiding the decision to instead proceed with resection via a right-sided transcavernous approach (Figure 3). The inability to sufficiently access the tumor through the anterior petrosectomy was corroborated intraoperatively. The transcavernous approach guided by the 3D segmentation helped facilitate a gross total resection (GTR), as confirmed by intraoperative MRI (Figure 4). The patient’s neurological examination was stable postoperatively, and her pathology was consistent with a grade I chondrosarcoma, thereby sparing her radiation therapy given her GTR. Her sixth nerve palsy entirely resolved at her six-month follow-up. Surveillance imaging at her one-year follow-up confirmed GTR and the absence of recurrence. The detailed surgical techniques employed during this case have been recently published [7].

#### 3.5.2. Case 2

A 64-year-old female (patient # 8) presented with recurrence of her clival chordoma five years after her initial transcondylar and occipito-C2 fusion surgery, followed by proton radiation treatment. She presented with worsening aspiration. Her MRI depicted a recurrence extending into the right cerebellopontine angle. She underwent surgery in the AMIGO suite after careful preoperative planning. Tumor resection was obtained through a restrictive window because of the presence of the fusion hardware and the position of the carotid and vertebral arteries, as well as the jugular bulb. The surgical cavity was then filled with harvested abdominal fat. Intraoperative T1 and contrast T1 with fat saturation as well as T2 images showed an island of residual tumor within the retropharyngeal area ventral to the clivus. The residual tumor was segmented, and the intraoperative images were co-registered to the initial surgical space. The updated navigation was employed to guide resection to the segmented area directly. After minimal drilling of residual normal-appearing condylar bone, the residual tumor was visualized, allowing its resection. Postoperative imaging was consistent with resection of the targeted tumor. The patient’s immediate postoperative exam was stable with progressive improvement of lower cranial nerve deficits over the following months (Appendix A).

## 4. Discussion

Surgical techniques and outcomes are constantly improving with technological advancements that have been introduced to the operating room over the past decades. Multiple imaging modalities are now available intraoperatively to provide visualization beyond what the surgeon can see directly. The integration of these modalities during surgery guides operative steps potentiating the benefits provided by each technique [8,9,10,11,12,13,14,15,16]. Our institution has established the Advanced Multimodality Image-Guided Operating (AMIGO) suite along this philosophy. The potential of this environment was shown to be invaluable in challenging surgeries and was implemented for intra-axial tumor and skull base surgery, increasing the safety and the extent of resection [3,10,17,18]. Achieving gross total resection during the first surgical intervention is particularly important with specific pathologies, specifically in chondrosarcomas and chordomas, as recurrences are frequently impossible to resect due to postsurgical and post-radiation scarring and vascular fragility [19,20,21,22]. Even though GTR of chordoma and chondrosarcoma is a critical factor in OS and PFS in chordomas and chondrosarcomas [4,5,23,24], achieving complete resection remains challenging, with gross total resection having been reported only in ~ 70% or less of patients [25,26,27]. Radical resection can be curative alone in grade I chondrosarcomas and can prolong survival in higher grade chondrosarcoma and chordomas after radiation [26,27,28,29]. If not achieved during the first surgical resection, especially with chordoma, radical resection becomes almost impossible due to extensive postsurgical and post-radiation effects, including radiation-induced angiopathy of intracranial vessels [4,23,26]. Hence, increased morbidity during recurrent tumor surgeries has been demonstrated [29]. When radical resection is unattainable, in radiated recurrent lesions, the surgical management becomes palliative. Maximal resection remains, however, central as it increases local control rates, improving PFS and OS in these patients [4,26,30]. This underlines the importance of multimodal imaging approaches in maximizing resection while decreasing complication rates [24].

High precision can be obtained in the skull base registration and can crucially be maintained throughout the case, as no shift occurs in the skull base during the resection, unlike during intra-axial surgery. Although visualization of the residual tumor via intraoperative imaging is critical, its co-registration to the prior surgical field, thereby allowing direct guidance to the targeted residual, is crucial [31]. As surgery on recurrent chordomas that have been radiated carries a high risk of vascular injuries due to the fragility of the radiated vessels in the field, the availability of endovascular intervention provides a crucial backup for safety and treatment. This vascular setup also allows fluoroscopy for the precise and safe placement of lumbar drains and fusion hardware when needed.

In our case series, the average extent of resection reached 97.7%, which was in line with that which was recently reported by Metwali et al. in their series focused on iMRI, endoscopic endonasal, and transoral approaches [32]. They also reported, as per our series, low surgical morbidity. Thus, regardless of the surgical route and the different patient population in the two series, the combined data speaks to the importance of the multimodal imaging and approach when resecting skull base chordomas and chondrosarcomas. We believe this multimodal approach combining intraoperative MRI and CT with ultrasounds, fluoroscopy, angiography, navigation, and endoscopic techniques enhances the benefit of each modality and yields better surgical outcomes in challenging cases as encountered in our series.

## 5. Limitations

Due to the rarity of chordomas and chondrosarcomas, our study involved a limited number of patients. This series contains a high rate of recurrent chordoma, which is associated with limited control and a higher complication rate. The availability of a surgical environment comparable to the AMIGO suite is clearly limited to specific institutions, and generalization cannot be easily made to most surgical environments. However, techniques such as intraoperative CT and MRI, ultrasounds, endoscopes, as well as neuronavigation with intraoperative co-registration, fluoroscopy, and endovascular interventions are becoming widely available, and their use is rapidly increasing as these technological tools become more affordable.

## 6. Conclusions

Advanced multimodality intraoperative imaging is valuable in the surgical management of skull base chordomas and chondrosarcomas and supports improved surgical outcomes. Immediate availability of intraoperative angiography and endovascular procedures can be vital, particularly during the resection of previously irradiated lesions. We find that intraoperative integration of the different imaging modalities via real-time navigation is pivotal to successful safe radical resection.

## Figures and Tables

**Figure 1 cancers-14-00966-f001:**
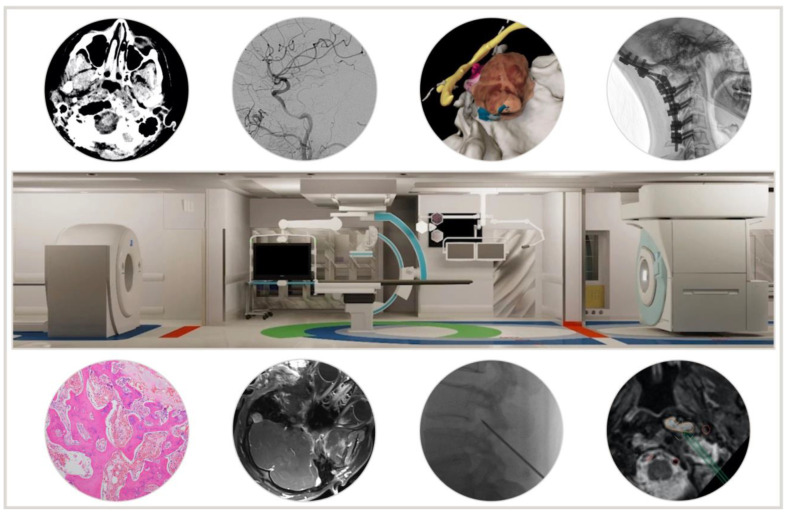
Advanced Multimodality Image-Guided Operating (AMIGO) suite and examples of the different modalities acquired and used in patients from this series.

**Figure 2 cancers-14-00966-f002:**
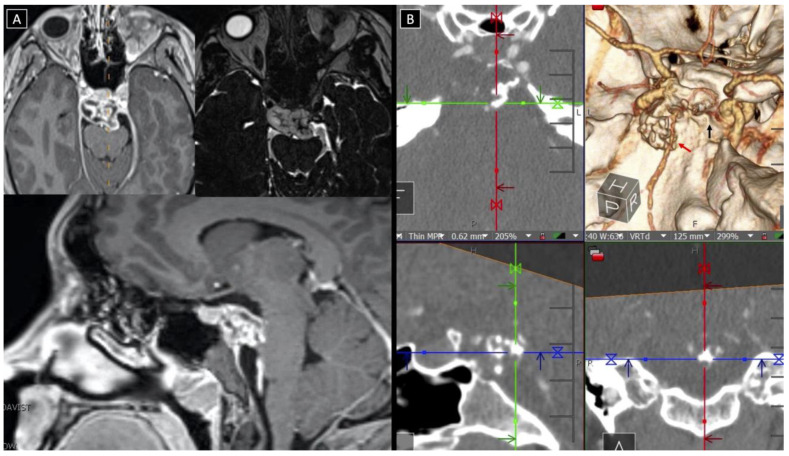
(**A**) MRI showering an upper clival contrast-enhancing lesion with mass effect on the brainstem and basilar artery and intradural extension, (**B**) CTA (axial, coronal, sagittal, and 3D reconstruction) showing the extent of intra-tumor calcifications, the proximity to the basilar system (red arrow), and the erosion of the posterior clinoid process (black arrow).

**Figure 3 cancers-14-00966-f003:**
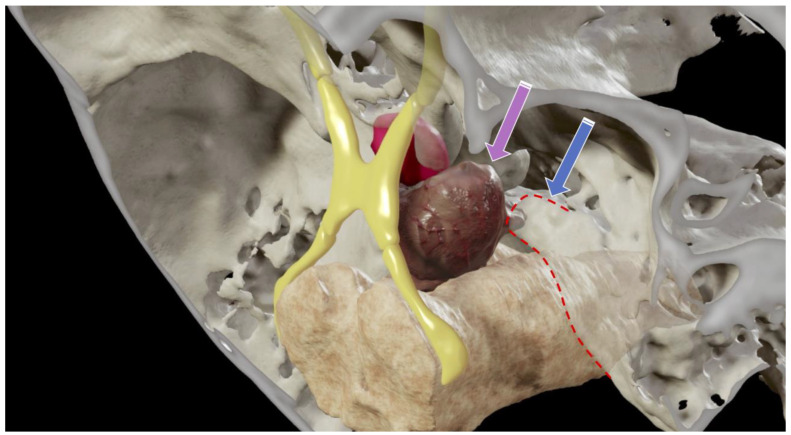
Three-dimensional reconstruction showing the potential surgical corridor provided by an anterior petrosal approach (blue arrow) versus the transcavernous approach (pink arrow). Optic apparatus (yellow), petrous apex and ridge limit (dotted red line), pituitary gland (magenta), tumor (maroon).

**Figure 4 cancers-14-00966-f004:**
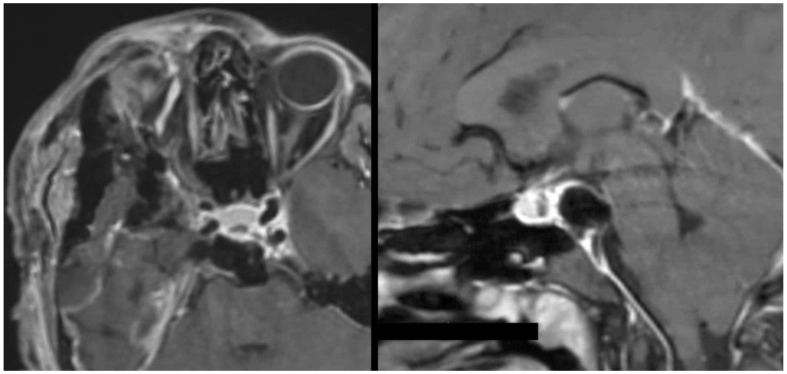
Intraoperative T1-weighted MRI with contrast and fat saturation showing gross total resection of the upper clival chondrosarcoma.

**Table 1 cancers-14-00966-t001:** Patient population.

Surg	Pts	Age	Sex	Symptoms	Tumor Location	Type of Tumor (Prior Surgeries)	Previous Radiation
1	1	71	F	Radiological progression	Rt CS	Recurrence (3)	Proton beam radiation
2	2	58	M	Increasing diplopia	Cl, PPS, Lt CS	Recurrence (2)	Proton beam radiation and Gamma knife
3	3	39	M	Increasing diplopia	Rt Cl, CS	Recurrence (1)	RT
4		42	M	Worsening vision in right eye	Rt Cl, MF, PPS, S, CS, NP	Recurrence (2)	RT
5		45	M	Radiological progression	Rt Cl, MF, PPS, S, CS, NP	Recurrence (3)	RT
6	4	55	M	Headaches, blurry vision	Cl, CS, S, PA	New	No
7	5	45	F	Right-sided jaw pain, neck pain	Cl, Bilateral condyles	New	No
8	6	45	F	Decreased hearing in the Rt	Rt PA, Cl	New	No
9	7	51	F	None	Rt PA, Cl	New	No
10	8	64	F	Worsening voice and swallowing	Cl, Bilateral condyles, retropharyngeal	Recurrence (1)	Proton beam radiation
11	9	24	F	Double vision	Upper Clivus	New	No

Cl: Clivus, CS: Cavernous sinus, F: Female, ITF: Infratemporal fossa, Lt: Left, M: Male; MF: Middle fossa, NP: Nasopharynx, PA: Petrous apex, PPS: Pre-Pontine space, Pts: Patients; Rt: Right, RT: Radiation therapy; S: Sella, Surg: Surgeries.

**Table 2 cancers-14-00966-t002:** Chief complaints, clinical findings, and tumor location.

Complaints				Clinical Findings				Features		Total	%
Neuro-Ophthalmologic			%	CN Deficit			%	Clival			
	Abnormal eye movement	5	45%		Optic	2	18%		Upper	9	82%
	Double vision	5	45%		Oculomotor	4	36%		Middle	9	82%
	Eye Drop	4	36%		Trochlear	2	18%		Lower	9	82%
	Visual difficulties	6	55%		Trigeminal	1	9%	Extension			0%
Headaches		2	18%		Abducens	5	45%		Cavernous sinus	8	73%
Neck pain		2	18%		Facial	0	0%		Sellar region	7	64%
Voice hoarseness		1	9%		Vestibulocochlear	1	9%		Intradural	5	45%
Swallowing difficulties		1	9%		Glossopharyngeal	1	9%		Petrous ridge	8	73%
Vertigo		1	9%		Vagus	1	9%		Cerebellopontine angle/prepontine	4	36%
					Accessory	0	0%		Jugular foramen	2	18%
					Hypoglossal	2	18%		Retro pharynx	4	36%
				Long tracts		1	9%		Infratemporal fossa	2	18%
				Cerebellar signs		0	0%		Occipital condyles	3	27%
									Cervical spine	1	9%
								Mean Tumor Volume (cm^3^)		13.4	SD 12

**Table 3 cancers-14-00966-t003:** Surgeries and outcomes.

Surg		Modalities	Modalities Used	Residual iMRI	Complications	Major Events in the Long Term Follow-Up
1	Right anterior preauricular	1	MRI	Anterior genu of the right carotid	None	No recurrence on the available one-year follow-up
2	Left preauricular and zygomatic	2	MRI, Endosc	Lt PPS, Lt CS, Post clinoid, Meckel’s cave	None	Multifocal disease including spinal cord, deceased
3	Right preauricular zygomatic middle fossa approach	4	MRI, CT, Endosc, Fluoro	Anterior CS	None	Local recurrence (S4)
4	Right preauricular zygomatic middle fossa approach	3	MRI, Endosc, Fluoro	Anterior CS	None	Local recurrence (S5)
5	Right preauricular zygomatic middle fossa approach	3	MRI, Endosc, Angio	Rt PPS, Anterior CS	Intracavernous carotid injury	Developed local recurrence and distal drop metastasis to the lumbar spine
6	Left preauricular middle fossa anterior petrosal approach (3 weeks after anterior transsphenoidal approach)	4	MRI, CT, Endosc, Fluoro	Petroclival/occipital clivus	Left V1-V2 hypoalgesia, L frontalis branch of facial	Radiation induced panhypopituitarism
7	Bilateral transcondylar approach with O-C5 fusion	3	MRI, Endosc, Fluoro	Base of the odontoid	DVT	Multifocal metastatic disease, including spine requiring radiation therapy, contralateral intradural infratentorial met requiring surgery in August 2020
8	Right anterior preauricular	3	MRI, Endosc, Fluoro		None	No recurrent disease on the 5 year follow-up
9	Right anterior preauricular	3	MRI, Endosc, Fluoro		Intracranial hypotension, Partial Rt VII	Initial progression of residual, stable after radiation therapy
10	Rt far lateral	2	MRI, Endosc	Retropharyngeal	None	Aggressive recurrence on 3–4 month postoperative imaging
11	R orbitozygomatic transcavernous	3	MRI, Endosc, Fluoro		None	No recurrence on the available one-year follow-up

Angio: angiography, CS: cavernous sinus, Endosc: endoscope, Fluoro: fluoroscopy, iMRI: intraoperative MRI, Lt: left, PPS: Pre-pontine space, Rt: right.

**Table 4 cancers-14-00966-t004:** Summary of resection, pathology, and follow-up results.

Extent of Resection		Patients	%
	GTR	2	18.2%
	STR/partial resection	9	81.8%
Pathology			
Chordoma	Classic	8	72.7%
	Chondroid	0	0.0%
	Dedifferentiated	0	0.0%
Chondrosarcoma	Grade I	3	27.3%
	Myxoid	2	18.1%
Symptoms/Deficit	Active	6	54.5%
Postoperative symptoms	Improvement	2	18.2%
	Stable	7	63.6%
	Worsening	1	9.1%
Morbidity	Major	1	9.1%
	Minor	4	36.4%
Follow-up		Years	SD
Progression-free survival (index surgery to clinical/rad progression)		2.3	2.3
Postoperative follow-up (index surgery to last available follow-up)		2.9	2.7
Overall follow-up (initial diagnosis-death/last follow-up in months)		11.48*	8.0

(*) Only one deceased patient, 22 years after initial diagnosis and 2 years following surgical intervention. GTR: gross total resection, STR: subtotal resection, SD: standard deviation.

**Table 5 cancers-14-00966-t005:** Volumetric analysis.

Surgery	1	2	3	4	5	6	7	8	9	10	11
Preop Volume (cm^3^)	3.78	16.82	7.15	10.28	25.84	22.23	40.74	2.56	2.56	10.03	5.50
iMRI volume (cm^3^)	0.13	0.13	0.60	0.28	2.74	0.11	2.88	0.00	0.44	1.02	0.00
Intraop % of reduction	96.6%	99.2%	91.6%	97.3%	89.4%	99.5%	92.9%	100.0%	82.8%	89.8%	100.0%
Postop MRI (cm^3^)	0.13	0.13	0.07	0.09	n/a	0.00	0.76	0.00	0.2	1.39	n/a
Postop % of reduction	96.6%	99.2%	99.0%	99.1%	89.4%	100.0%	98.1%	100.0%	92.2%	n/a *	100.0%

* No immediate postoperative imaging available, aggressive recurrence on the 3-month postoperative imaging.

## Data Availability

Not applicable.

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
