# Peer review of "Multimodal Intraoperative Image-Driven Surgery for Skull Base Chordomas and Chondrosarcomas"

_cancers, 2022, doi:10.3390/cancers14040966_

Round 1

Reviewer 1 Report

The authors report a series of nine patients undergoing 11 surgeries for newly diagnosed or recurrent chordoma. The authors describe their surgical approach using a multimodality image guided operating suite.  As we all know surgery of chordomas, especially of recurrent chordomas, is challenging and the authors should be congratulated for their thoroughly surgical planning and approaches, using elaborate pre- and intraoperative imaging, which is well described throughout the manuscript. The number of patients presented is low. However, chordoma is a rare disease and reports on large patient series are rare. However, patients’ symptoms, tumour location, surgical approach, imaging, complications and outcome are well documented in the tables, and additionally illustrated by two case reports. There is not much to add and publication can be recommended.

Comments

How is subtotal resection defined in chordoma (% of volume remaining?)

What means Volume reduction 0 and 0% in Table 4?

What means Pathology 0 in table 4?      

Author Response

Reviewer 1

The authors report a series of nine patients undergoing 11 surgeries for newly diagnosed or recurrent chordoma. The authors describe their surgical approach using a multimodality image guided operating suite.  As we all know surgery of chordomas, especially of recurrent chordomas, is challenging and the authors should be congratulated for their thoroughly surgical planning and approaches, using elaborate pre- and intraoperative imaging, which is well described throughout the manuscript. The number of patients presented is low. However, chordoma is a rare disease and reports on large patient series are rare. However, patients’ symptoms, tumour location, surgical approach, imaging, complications and outcome are well documented in the tables, and additionally illustrated by two case reports. There is not much to add and publication can be recommended.

Comments

  • How is subtotal resection defined in chordoma (% of volume remaining?)
  • What means Volume reduction 0 and 0% in Table 4?
  • What means Pathology 0 in table 4?

Response

We are grateful to the reviewer for their favorable review and constructive input

1-Subtotal resection is usually defined as a resection of 95-90%, with <90% defined as partial. We addended the submission in line 93-95 as well as table 4 as recommended.

2- and 3- : We are grateful to the reviewer for catching these mistakes. Both volume reduction and pathology associated numbers are formatting errors and we addended the table to correct these mistakes.

Reviewer 2 Report

Muliti-modality intraoperative imaging is an important adjunct for the neurosurgeon and in this regard, the paper is valuable.  The authors are to be commended for pursing this approach in a small series of patients with difficult diagnoses.

I have several concerns about the paper.  First, when one reads it initially it would appear that there are many different modalities available for intraoperative imaging.  

However, the PET scan does not appear to have actually been used for intraoperative repeat imaging to assess resection, and fluoroscopy and 3D reconstructions using Brain Lab software are standard fare in many operating rooms.  Intraoperative angiography might have been used in the case of vascular injury but this also is available in many institutions. 

The authors appear to really only use CT and MRI intraoperatively, with the other modalities imported preoperatively into the Brain Lab navigational system.  3D reconstructions of newly acquired data is a capability of the software, and is available in many places.  

Thus it would be helpful for the authors to clarify this.  The AMIGO suite has great capabilities, but when it comes to intraoperative imaging, it is CT and MRI scans that are made, and then using Brain Lab and possibly other software, that information is processed and redisplayed including 3D reconstructions.  Making this clear is important for others who either use the technique or who want to do so.  All that is needed is easy intra-operative access to CT and MR imaging and the ability to import and reconstruct the data real time.  That is plenty and a challenge for many, but the other imaging modalities used are either standard fare in many centers (x-ray imaging, endoscopy) or do not need to be in adjacent rooms (PET scanning and angiography (can be done in the primary Or).

This does not detract from the value of the technique, but it is important to clarify that what really is happening is a variety of preoperative imaging is imported and co-registered into the Brain Lab, and then intraoperative MR and occasionally CT data is again obtained and processed and analyzed.

The results, while admirable, cannot easily be compared to other series, as the numbers are small and each case has its own complex series of characteristics that make it unique.  It is therefore difficult to state that  the results were clearly improved or to even compare to other series, without larger numbers and direct comparison or classification of individual tumor locations and characteristics in differing series. 

With these caveats clarified in a modified version, the paper is valuable and could be published. 

Author Response

Reviewer 2

Multi-modality intraoperative imaging is an important adjunct for the neurosurgeon and in this regard, the paper is valuable.  The authors are to be commended for pursing this approach in a small series of patients with difficult diagnoses.

I have several concerns about the paper. 

  • First, when one reads it initially it would appear that there are many different modalities available for intraoperative imaging. However, the PET scan does not appear to have actually been used for intraoperative repeat imaging to assess resection, and fluoroscopy and 3D reconstructions using Brain Lab software are standard fare in many operating rooms.  Intraoperative angiography might have been used in the case of vascular injury but this also is available in many institutions. The authors appear to really only use CT and MRI intraoperatively, with the other modalities imported preoperatively into the Brain Lab navigational system. 3D reconstructions of newly acquired data is a capability of the software, and is available in many places.  Thus it would be helpful for the authors to clarify this.  The AMIGO suite has great capabilities, but when it comes to intraoperative imaging, it is CT and MRI scans that are made, and then using Brain Lab and possibly other software, that information is processed and redisplayed including 3D reconstructions.  Making this clear is important for others who either use the technique or who want to do so.  All that is needed is easy intra-operative access to CT and MR imaging and the ability to import and reconstruct the data real time.  That is plenty and a challenge for many, but the other imaging modalities used are either standard fare in many centers (x-ray imaging, endoscopy) or do not need to be in adjacent rooms (PET scanning and angiography (can be done in the primary Or).

  • This does not detract from the value of the technique, but it is important to clarify that what really is happening is a variety of preoperative imaging is imported and co-registered into the Brain Lab, and then intraoperative MR and occasionally CT data is again obtained and processed and analyzed.

  • The results, while admirable, cannot easily be compared to other series, as the numbers are small and each case has its own complex series of characteristics that make it unique.  It is therefore difficult to state that the results were clearly improved or to even compare to other series, without larger numbers and direct comparison or classification of individual tumor locations and characteristics in differing series. 

With these caveats clarified in a modified version, the paper is valuable and could be published. 

Response

We are grateful to the reviewer for their favorable review and constructive comments. We addended our submission according to the suggestions as follows:

  • We agree with the reviewer that useful tools and techniques are available in different modalities and certainly surgeons dealing with various tumors utilize these as needed. We are fortunate to have access to this multi-modality image guided operating suite, and here we report on its use in specific rare tumors, chordomas and chondrosarcomas. We agree that the use of the PET-CT has no use in these cases, and we mentioned it only as part of the general introduction of the AMIGO suite. We addended the manuscript (line 260-264) as recommended

  • We agree with the reviewer that it is the acquisition of multimodality (CT and MRI), with registration of preoperative imaging then intraoperative co-registration. We shared these steps in the “Operative and perioperative management” section (line 60).

  • As acknowledged in the limitation section and noted by the reviewer, the rarity of these tumors and the small number of patients in most series, including ours, limits analysis. Hence, we did not aim to generalize our outcome. Our focus was on how the use of these modern tools can facilitate maximum safe tumor removal which is considered a pivotal factor in the management of chordoma and chondrosarcoma.